# Improving Neural Network Accuracy by Concurrently Training with a Twin Network

**Benjamin Vandersmissen**[*]
University of Antwerp, sqIRL/IDLab, imec
`benjamin.vandersmissen@uantwerpen.be`

**Lucas Deckers**[*]
University of Antwerp, IDLab, imec

**Jose Oramas**
University of Antwerp, sqIRL/IDLab, imec

## Abstract

Recently within Spiking Neural Networks, a method called Twin Network Augmentation (TNA) has been introduced. This technique claims to improve the validation accuracy of a Spiking Neural Network simply by training two networks in conjunction and matching the logits via the Mean Squared Error loss. In this paper, we validate the viability of this method on a wide range of popular Convolutional Neural Network (CNN) benchmarks and compare this approach to existing Knowledge Distillation schemes. Next, we conduct an in-depth study of the different components that make up TNA and determine that its effectiveness is not solely situated in an increase of trainable parameters, but rather the effect of the training methodology. Finally, we analyse the representations learned by networks trained with TNA and highlight their superiority in a number of tasks, thus proving empirically the applicability of Twin Network Augmentation on CNN models.

## 1 Introduction

Training a neural network typically follows a standard setup where it is optimized via an objective function to align the predictions for a set of inputs to a corresponding set of targets. During this procedure, the internal layers of the neural network learn to encode informative features that assist in the accurate prediction of the target labels. To further increase the quality and usefulness of those internal features, different approaches exist such as regularization to prevent overfitting (Ng, 2004; Hinton, 2012), and data augmentation (Cubuk et al., 2019; 2020) to increase input variety.

Beyond aiding in classification, the learned features of the model can have numerous other uses. For example, it has been demonstrated that internal features can be used to cluster and retrieve semantically similar inputs (Babenko et al., 2014), accelerate the training of smaller models via Knowledge Distillation (KD) (Hinton et al., 2015) or Deep Mutual Learning (DML) (Zhang et al., 2018), or even allow the same model to generalize on a different dataset (Weiss et al., 2016). These show the importance of learning strong, usable features, which we also explore in this work.

Recently, Deckers et al. (2024) introduced a method called Twin Network Augmentation (TNA) which involves optimizing two Spiking Neural Networks (SNNs) concurrently on a classification task and aligning their logits during training via an additional Mean Squared Error (MSE) loss term, which results in greatly improved single network test accuracy. While this approach has been shown successfully in the context of SNNs, it has however not been proven that this methodology is applicable across other non-spiking types of neural networks and other different model architectures of machine learning in general.

To this end, we explore the effects of this approach on standard Artificial Neural Networkss (ANNs), and more specifically, on a range of benchmark CNN models. Worth-noting is that different from the stateless neuron models that compose standard ANNs; typically characterized by weights, biases and an activation function, such as the ReLU, a spiking neuron integrates its inputs over time

---

[*]Denotes equal contribution

and generates binary events depending on its weights, biases and neuronal thresholds. As this is a substantial difference, and considering the wide usage of standard ANNs, we strive to determine whether TNA is a valid training strategy for CNNs.

Our contributions are the following.

1. We show the applicability of the recently introduced Twin Network Augmentation (TNA) (Deckers et al., 2024) on Convolutional Neural Networks (CNNs).

2. We corroborate on the superiority of L2 matching the logits w.r.t. using Kullback-Leibler (KL) Divergence on the probabilities (Kim et al., 2021) in the Online KD setting.

3. We demonstrate that the increased effectiveness of TNA cannot naively be reduced to an increase in trainable parameters, but rather is related to both network reinforcing each other's predictions, meaning the whole is greater than the sum of its parts.

4. Via an in-depth analysis, we determine that the features learned via TNA exhibit greater predictive capabilities at most levels and show a greater robustness to data corruptions.

The rest of the paper is structured as follows. In section 2 we highlight related methods and approaches. In section 3 we reintroduce Twin Network Augmentation (TNA). Via experiments, we show the impact of applying TNA in section 4, and we conclude this work in section 5.

## 2 RELATED WORK

### 2.1 KNOWLEDGE DISTILLATION

**Pretrained Teacher.** Knowledge Distillation (KD) aims to exploit the learned features within a fully trained neural network (a 'teacher network') to accelerate, and improve the training process of a secondary network (a 'student network'). A typical approach uses a combination of a task-specific loss such as the Cross-Entropy loss for classification, and a distillation loss which allows the student to learn from the teacher. The foundational approach (Hinton et al., 2015) introduces the notion of soft targets, which are the output probabilities of a teacher network given an input, and argue that these soft targets provide more nuanced views of the input compared to hard targets, namely ground-truth labels. They introduce a distillation loss based on the Kullback-Leibler (KL) Divergence (Kullback & Leibler, 1951) such that the student model learns to mimic the probability distribution of the teacher model. Kim et al. (2021) studies the effect of the temperature parameter $\tau$ in the KL divergence and finds that a high $\tau$ is equivalent to logit matching, while a $\tau$ close to 0 is equivalent to label matching. They furthermore derive a relationship between KL divergence with a large $\tau$ and the MSE loss on the logits. A secondary approach (Romero et al., 2015), proposed to match hidden activations, thus allowing the student network to learn similar features to the teacher network. A generalization of the previous approaches is Relational KD (Park et al., 2019). Rather than matching the knowledge extracted from single samples, instead relations between the knowledge of multiple samples is matched. This can be done in a distance-based manner (second-order matching) or in an angle-based manner (third-order matching). Another approach that has shown promise is self-distillation (Zhang et al., 2019), in rather than viewing a model as a monolith, it is instead considered as a stack of submodels, where each submodule feeds in one another. Self disitillation then uses the submodels as students and the full model as teacher. Then, the submodels are updated concurrently with a mix of KD losses, while the full model is training.

**Online Knowledge Distillation and Deep Mutual Learning.** Rather than using a fixed pretrained teacher network, Online Knowledge Distillation (Guo et al., 2020) allows for the teacher model to evolve during its operation. This is useful in cases where the data distribution changes throughout time, and allows the teacher to include the new data distribution within its learned features. Deep Mutual Learning (DML) (Zhang et al., 2018) goes a step further, rather than using a teacher network, instead a set of student networks – called a cohort – are trained concurrently. Typically each student model is trained on hard targets via the cross entropy loss and additionally, on soft targets using the KL Divergence Loss.

**Comparison with TNA.** While DML is the field that lies the closest to TNA, the key difference is that TNA works directly on the logits of the model, rather than the SoftMax probabilities. This allows us to use a different loss function, namely the MSE Loss, compared to the KL Divergence Loss,

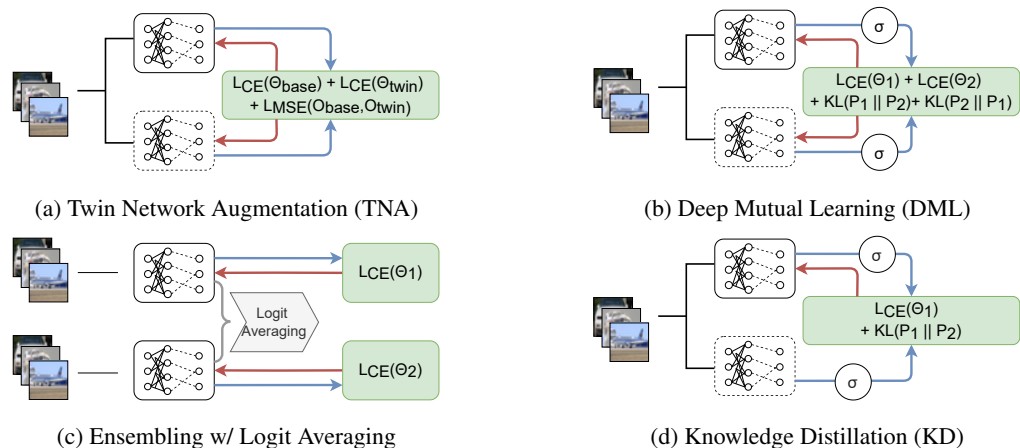

Figure 1: A comparison of TNA to different related approaches. Note that Ensembling employs both models during inference and KD relies on a frozen pretrained network during training. TNA focuses on matching logits, while DML and KD focus on matching probabilities.

which is only applicable on probability distributions. We state that the MSE-based logit matching allows for a more direct comparison and therefore leads to better generalized latent representations. See fig. 1 for a visual comparison.

## 2.2 SIAMESE NETWORKS AND CONTRASTIVE LEARNING

Siamese networks (Chicco, 2021) consist of two (or more) identical neural networks that are trained concurrently. Different from TNA, these networks are trained in a contrastive fashion, with a loss such as the Triplet Loss (Schultz & Joachims, 2003), and each weight update is applied on all the networks involved, thus all networks have always identical weights. In contrast, TNA starts with differently initialized networks, employs a cross-entropy loss to learn for classification, and has separate weight updates per network.

## 2.3 ENSEMBLING AND MIXTURE OF EXPERTS

Ensembling (Ju et al., 2018) is a technique which aggregates the predictions of multiple fully trained networks, either via a voting system or via averaging the individual logits. By ensembling, it is possible to produce a classifier that outperforms an individual model, at the cost of additional parameters during inference. Mixture of Experts (MoE) (Jacobs et al., 1991) is similar in this regard, but rather than having classifiers that can recognize all classes, instead the involved models are specialised in a limited number of classes. The MoE structure includes a routing mechanism to efficiently determine to which experts a specific input should be fed to. In contrast, TNA also involves multiple models that are trained, however while in Ensembling and MoE these models are trained individually, in TNA these models are trained in an entangled manner. Additionally, during inference, we only use a single model, different to the previously-mentioned methods, which leads to reduced inference costs when compared to those methods.

## 3 METHODOLOGY

### 3.1 TWIN NETWORK AUGMENTATION

Contrastive Learning is a technique that employs augmented views of the same datapoint to learn cohesive usable representations. This is done by employing two distinct transformation functions and feeding both through an encoder network which computes representations. The two representations are transformed via a projection head onto a different space, on which the contrastive loss is applied with the aim of steering feature similarity. After training, then the projection head is discarded.

We hypothesise that a similar process can be exploited using different (partially) trained neural networks. Rather than generating different views of a datapoint via augmentations, we can feed the same sample through different networks. As both networks have different weights, the views they provide are different from one another. Additionally, if we assume that both networks are similarly trained, i.e., have a similar validation accuracy, then those views should be of similar quality.

We use this concept to reintroduce Twin Network Augmentation (TNA) from a representation learning perspective. TNA is a training paradigm that trains multiple networks in parallel with the goal of improving the accuracy of the base network. To achieve this, in conjunction with the standard Cross-Entropy losses on each model, an additional Mean Squared Error loss is applied on the logits of both networks. The goal of this term is to force the outputs of the networks to grow closer and incorporate the information of the multiple views of the datapoint in the training process. Having multiple views of the same input allows the model to learn more robust features and reduces the risk of overfitting on certain data points.

Formally, TNA is defined as follows. Given two networks, a base network $F_A$ and a twin network $F_B$ with weights $\Theta_A$ and $\Theta_B$ respectively, we define the output logits of $F_A$ given an input $\mathcal{I}$ as $F_A(\mathcal{I}) = o_A$. This leads us to the following definition of the TNA training procedure.

$$\mathcal{L}_{\text{TNA}} = \underbrace{\mathcal{L}_{\text{CE}}^A}_{\text{Base Loss}} + \underbrace{\mathcal{L}_{\text{CE}}^B}_{\text{Twin Loss}} + \alpha \cdot \underbrace{||o_A - o_B||_2^2}_{\text{Logit Matching Loss}} \tag{1}$$

During inference, we can discard the twin network $F_B$, thus only using the base network $F_A$ to make predictions. This means that no additional overhead is incurred when making predictions. An additional option would be to ensemble both models for a minor performance gain at the cost of a longer inference time.

## 3.2 Comparison with Mutual Learning

The loss of network $F_A$ trained with Deep Mutual Learning (DML) (Zhang et al., 2018) on a dataset with $M$ classes and $N$ samples can be defined as follows:

$$\mathcal{L}_{\text{ML}}^A = \mathcal{L}_{\text{CE}}^A + \mathcal{D}_{\text{KL}}(p_B || p_A) \tag{2}$$

with $\mathcal{D}_{\text{KL}}(p_B || p_A)$ the Kullback-Leibler (KL) Divergence as defined below.

$$\mathcal{D}_{\text{KL}}(p_B || p_A) = \sum_{i=1}^{N} \sum_{j=1}^{M} p_B^j(x_i) \log \frac{p_B^m(x_i)}{p_A^m(x_i)} \tag{3}$$

with $p_A^m(x_i)$ the probability of class $m$ for network $F_A$ given a sample $x_i$; computed via SoftMax

$$p_A^m(x_i) = \frac{exp(o_A{}^m)}{\sum_{m=1}^{M} exp(o_A{}^m)} \tag{4}$$

Different to DML, we do not aim to match probability distributions, but rather the logits. We argue that matching a probability distribution, which is generated by applying SoftMax on the logits, discards useful information due to the rescaling properties of SoftMax. We base ourselves on the work by Kim et al. (2021) that shows that minimizing KL divergence loss with a temperature $\tau \to \infty$ is equivalent to minimizing MSE loss with an additional constant. By following the assumption that the teacher's logit mean is 0, this constant forces the student logit mean to diverge from zero. The authors also prove this empirically.

## 3.3 Experimental Setup

We apply the TNA methodology on different baselines from the Computer Vision domain, specifically focusing on CNNs. We use a simple ConvNet, a ResNet-18 (He et al., 2016), and a MobileNetv2 (Sandler et al., 2018) on datasets such CIFAR-10 and CIFAR-100 (Krizhevsky et al.,

Table 1: Results of applying TNA on the different dataset network combinations.

| Network | Dataset | Single Train | TNA | Gain |
|---------|---------|--------------|-----|------|
| ConvNet | CIFAR-10 | $86.84 \pm 0.14\%$ | $87.07 \pm 0.31\%$ | +0.23% |
| ResNet-18 | CIFAR-10 | $95.06 \pm 0.05\%$ | $95.38 \pm 0.18\%$ | +0.32% |
| MobileNetv2 | CIFAR-10 | $93.94 \pm 0.22\%$ | $94.56 \pm 0.32\%$ | +0.62% |
| ConvNet | CIFAR-100 | $61.14 \pm 0.37\%$ | $61.83 \pm 0.36\%$ | +0.69% |
| ResNet-18 | CIFAR-100 | $77.66 \pm 0.28\%$ | $79.00 \pm 0.35\%$ | +1.34% |
| MobileNetv2 | CIFAR-100 | $76.17 \pm 0.15\%$ | $79.50 \pm 0.27\%$ | +3.33% |
| ConvNet | TinyImageNet | $37.75 \pm 0.23\%$ | $38.12 \pm 0.41\%$ | +0.37% |
| ResNet-18 | TinyImageNet | $59.33 \pm 0.57\%$ | $64.84 \pm 0.56\%$ | +5.51% |
| MobileNetv2 | TinyImageNet | $60.10 \pm 0.20\%$ | $65.11 \pm 0.09\%$ | +5.01% |
| ResNet-50 | ImageNet | $76.61 \pm 0.07\%$ | $77.65 \pm 0.01\%$ | +1.04% |
| ViT-S/16 | ImageNet | $69.83 \pm 0.12\%$ | $71.76 \pm 0.35\%$ | +1.97% |

Table 2: The performance of TNA compared to DML.

| Network | Dataset | lr | Single Train | TNA | DML |
|---------|---------|-----|--------------|-----|-----|
| MobileNet-v2 | CIFAR-100 | 0.1 | $74.74 \pm 0.16\%$ | $\mathbf{77.78 \pm 0.15\%}$ | $1.66 \pm 1.15\%$ |
| MobileNet-v2 | CIFAR-100 | 0.05 | $76.17 \pm 0.15\%$ | $\mathbf{79.50 \pm 0.27\%}$ | $78.08 \pm 0.34\%$ |
| ResNet-18 | TinyImageNet | 0.1 | $59.33 \pm 0.57\%$ | $\mathbf{64.84 \pm 0.56\%}$ | $0.05 \pm 0.00\%$ |
| ResNet-18 | TinyImageNet | 0.05 | $57.79 \pm 0.20\%$ | $\mathbf{64.34 \pm 0.29\%}$ | $62.98 \pm 0.44\%$ |

2009), and TinyImageNet (Le & Yang, 2015). More complex scenarios are considered with ResNet-50 (He et al., 2016), and ViT-S/16 (Beyer et al., 2022) on ImageNet (Deng et al., 2009). The full training configurations, as well as the architecture of the ConvNet are listed in the appendix. For each configuration, we apply Stochastic Gradient Descent, and use Cosine Learning Rate annealing during the training process. To ensure that our observations are representative of the trends, each experiment is repeated over three random runs.

# 4 EXPERIMENTS

## 4.1 TWIN NETWORK AUGMENTATION

By applying the Twin Network Augmentation methodology on different baselines, we notice a consistent increase in accuracy w.r.t. training a single network in table 1, thus showing that application of TNA is valid. However, for the ConvNet architecture, this increase is not alwas statistically significant. We link this to the limited capacity of the network, and hypothesize that this limits feature space exploration, and as such the performance of TNA. As DML follows a similar methodology, we compare with that approach in table 2. We notice that while networks trained with DML also significantly outperforms the single trained network, applying TNA results in an additional performance gain. It should be noted that in order for DML to converge to a good solution, we had to tweak the learning rate. In the case of MobileNet-v2 + CIFAR-100 this tweaking also had a positive effect on the other settings, while in the case of ResNet-18 + TinyImageNet, this negatively affects the baseline. This shows additionally that TNA is more robust to different values of hyperparameters. Next we strive to understand why this methodology improves validation accuracy.

## 4.2 TNA AND NETWORK CAPACITY.

One hypothesis for the increased performance of networks trained with TNA is that the effective capacity of the base network is artificially increased by virtue of the Logit Matching Loss. The reasoning behind this is that when two randomly initialized neural networks are trained separately, they converge to different solutions, encoding different features. By adding the Logit Matching Loss, the output of both networks is constrained to be similar, i.e., both networks will converge to

Table 3: Ensembling of differently trained ResNet-18 models on TinyImageNet. * indicates networks trained with lr=0.05 rather than lr=0.1.

|        | No Ensemble | Ensemble |
|--------|-------------|----------|
| Single | $59.36 \pm 0.57\%$ | $63.79 \pm 0.26\%$ |
| TNA    | $64.84 \pm 0.56\%$ | $\mathbf{66.56 \pm 0.24\%}$ |
| DML*   | $62.98 \pm 0.44\%$ | $65.04 \pm 0.03\%$ |

Table 4: Wide ResNet-18 performance with different width multipliers compared to a ResNet-18 trained with TNA on different datasets.

| Dataset | Width | | | TNA |
|---------|-------|------|------|-----|
|         | $1\times$ | $2\times$ | $4\times$ | |
| CIFAR100 | 77.56% | 79.12% | **79.22%** | 79.00% |
| TinyImageNet | 59.08% | 60.51% | 60.90% | **64.65%** |

a closely similar solution, but by using different initializations, both networks will explore different paths in the feature space. These different paths provide positive reinforcement for each other and lead to a general increase in performance.

To test this hypothesis, we conduct three tests. First, we compare TNA with a naive method to improve the capacity of a neural network, namely by training a single network with a modified width (number of filters of a network), thus leading to the same or higher number of trainable parameters. Second, we compare TNA with ensembling of regularly trained models, which is a technique that increases the capacity of a model during inference by averaging predictions of different models. Third, we conduct a progressive reinitialization test, starting from an identical twin and base network, to determine the impact of twin initialization on the TNA algorithm.

**Comparison with wider networks.** In table 4, we compare the performance of ResNet-18 with TNA to the training of different variants of single WideResNet-18 models. These are variations of ResNet-18 with different capacities, obtained by multiplying the number of filters in each layer by a specific factor. In this experiment we choose the multipliers $w \in \{1, 2, 4\}$, with w=1 equivalent to ResNet-18. In the case of CIFAR-100, the validation accuracy of the wider networks and the version with TNA lie well within a standard deviation in both directions, showing no difference of significance. In the case of TinyImageNet, the difference is much more pronounced, with TNA significantly outperforming the wider networks. This shows that the gain of TNA lies not only in a simple increase of the trainable parameters.

**Comparison with model ensembling.** Next, we study a more advanced approach of increasing the model capacity via model ensembling. We focus on logit averaging, which averages the outputs of a set of trained networks to get a single output. In the case of $k = 2$ models, this involves double the computational resources during both training and inference compared to training a single network. As such, it has a similar computational cost during training as the TNA technique, but double the inference cost. In table 3, we compare the performance of differently trained ResNet-18 models with that of an ensemble of two ResNet-18 models. We notice that TNA outperforms the ensemble of two standard ResNet-18s, which shows the additional efficiency that is encoded within the single network. Remarkably, even though the objective of TNA is to minimize the distance between logits of the base and the twin network, we still achieve a substantial accuracy increase when ensembling the base and the twin network with logit averaging. This suggests that our hypothesis that the twin architecture exploring multiple solutions in the feature space might be correct.

**Progressive Reinitialization.** A straight-forward assumption to make about TNA is that both networks need to be initialized differently. Of course, if both networks are totally similar at initialization, each network will produce the same logits given the same input and as such, the matching loss will be zero. However, to determine exactly how different the networks need to be for TNA to yield a noticeable increase in validation accuracy, we attempted TNA with a twin network that was only initialized differently in one layer. We noticed that no matter which layer we initialized differently, this had no significant impact on the validation accuracy, thus showing that initialization of the twin network doesn't matter for the performance of TNA.

### 4.3 FEATURES LEARNED BY TNA

Next we want to assess the quality of the features learned by TNA. In the previous section, we discussed that application of TNA can lead to increased validation accuracy on several benchmarks, but this still leaves a number of questions unanswered, such as: (1) How different are the learned

Table 5: Feature Similarity between ResNet-18 models trained on TinyImageNet with different approaches, measured with CKA. Both base and twin networks are trained concurrently with TNA, while DML A and DML B are trained concurrently with DML.

|         | Single vs Single     | TNA vs Single        | Base vs Twin         | DML A vs DML B       |
| ------- | -------------------- | -------------------- | -------------------- | -------------------- |
| Block 1 | $0.5201 \pm 0.3226$  | $0.7860 \pm 0.2863$  | $0.9798 \pm 0.0132$  | $0.9554 \pm 0.0231$  |
| Block 2 | $0.8758 \pm 0.0597$  | $0.9262 \pm 0.0714$  | $0.9660 \pm 0.0209$  | $0.9693 \pm 0.0100$  |
| Block 3 | $0.9292 \pm 0.0251$  | $0.9541 \pm 0.0169$  | $0.9770 \pm 0.0147$  | $0.9821 \pm 0.0044$  |
| Block 4 | $0.8651 \pm 0.0260$  | $0.8496 \pm 0.0650$  | $0.9749 \pm 0.0015$  | $0.9384 \pm 0.0102$  |
| Block 5 | $0.9493 \pm 0.0058$  | $0.9427 \pm 0.0158$  | $0.9823 \pm 0.0007$  | $0.9774 \pm 0.0002$  |
| Block 6 | $0.8752 \pm 0.0327$  | $0.8527 \pm 0.0426$  | $0.9442 \pm 0.0026$  | $0.9208 \pm 0.0024$  |
| Block 7 | $0.8746 \pm 0.0056$  | $0.8664 \pm 0.0095$  | $0.9349 \pm 0.0010$  | $0.9128 \pm 0.0024$  |
| Block 8 | $0.6683 \pm 0.0016$  | $0.6392 \pm 0.0029$  | $0.8132 \pm 0.0023$  | $0.7935 \pm 0.0020$  |
| FC      | $0.8556 \pm 0.0009$  | $0.8630 \pm 0.0026$  | $0.9714 \pm 0.0001$  | $0.9268 \pm 0.0002$  |

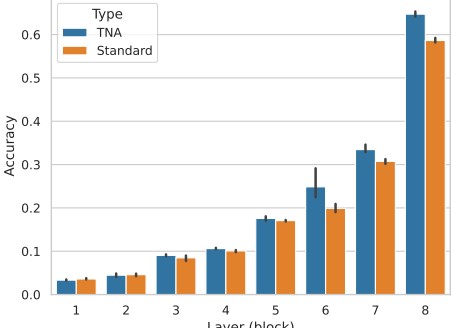

Figure 2: Validation accuracy of linear probes inserted at different points in a ResNet-18 trained on TinyImageNet.

Figure 3: Convergence behaviour of ResNet models at different sizes while training on TinyImageNet.

features from those encoded in single trained networks? (2) Are only the deeper-level features better, or is there a distinction already in the shallow-level features? (3) How much difference between the twin and base initialization is required for TNA to work?

**Feature Similarity.** We use the Centered Kernel Alignment (CKA) metric introduced by Kornblith et al. (2019) to directly compare features learned by different networks. Two experimental settings are considered, (1) a comparison of networks trained using various methods and (2) a comparison of networks learned in conjunction via DML or TNA. For each setting, we calculate the representation similarity with CKA at the end of each convolutional block in a ResNet-18 over the TinyImageNet validation set, and list the results in table 5. We notice that applying TNA results in increased feature similarity at each point in the network w.r.t. the baseline. While this is expected at the logit level – after the fully connected layer – as these are directly matched, this shows that both networks try to mimic each others features. Interestingly, we still notice a considerably larger difference in the high-level features, albeit less pronounced than in the case of two separately trained network, showing that these networks still consider different information. This indicates that the networks still consider different kinds of information to achieve similar logits. Additionally, we notice that the feature similarity of TNA trained networks is higher at the logit level (FC layer) than that of DML-trained networks, which indicates better agreement between the networks. Coupled with the results on classification benchmarks, this indicates that the better similarity of the logits achieved by TNA leads to a gain in classification accuracy.

**Intermediate feature quality.** While it is evident that the features at the last convolutional layer of TNA learned networks are better suited for classification, we next want to determine how discriminative the intermediate representations are. For this purpose we use a linear probing technique introduced by Alain & Bengio (2017). At certain points in the network, we train a linear classification head on the frozen weights of the preceding layers. To avoid large computations, we first

Table 6: ResNet-18 accuracy on different corruptions of the TinyImageNet-C dataset.

| Corruption | Single | TNA | Corruption | Single | TNA |
|---|---|---|---|---|---|
| Brightness | 36.35% | 42.45% | Impulse Noise | 24.09% | 25.29% |
| Contrast | 11.85% | 14.57% | JPEG Compression | 47.53% | 52.33% |
| Defocus blur | 26.57% | 28.60% | Motion Blur | 32.16% | 35.62% |
| Elastic Transform | 38.95% | 43.16% | Pixelate | 37.65% | 41.87% |
| Fog | 27.29% | 31.69% | Shot Noise | 25.09% | 27.56% |
| Frost | 29.69% | 31.69% | Snow | 27.09% | 32.53% |
| Gaussian Noise | 20.82% | 22.71% | Zoom Blur | 31.02% | 33.92% |
| Glass Blur | 21.99% | 23.32% | | | |
| Average | 29.88% | 32.49% | *None* | 59.33% | 64.84% |

Table 7: Ablation test of different values for the parameter $\alpha$. Entries indicated with a dash (-) did not converge and remained at random chance.

| Network | Dataset | $\alpha$ | | | | | | | |
|---|---|---|---|---|---|---|---|---|---|
| | | 0.0 | 1E-4 | 5E-4 | 1E-3 | 5E-3 | 1E-2 | 5E-2 | 1E-1 |
| ResNet-18 | TinyImageNet | 59.33% | 60.98% | 62.33% | 63.33% | **64.65%** | 63.79% | - | - |
| ResNet-18 | CIFAR-10 | 95.06% | 95.23% | 95.25% | **95.43%** | 95.38% | 95.33% | - | - |
| MobileNetv2 | CIFAR-100 | 76.17% | 76.54% | 77.12% | 77.43% | 77.86% | 78.61% | **79.50%** | 77.91% |

apply a dimensionality reduction in the form of Average Pooling across each channel. We show the results in fig. 2. We can see that the main accuracy gain of the networks trained with TNA comes from the deeper features. This lies in line with the results from the CKA analysis, thus showing that the reason for a better performance is a better combination of similar low-level features, resulting in better performing high-level features.

**Corruption Robustness.** Introduced by Hendrycks & Dietterich (2019), TinyImageNet-C is a dataset that consists of different perturbations applied on the TinyImageNet dataset. The goal of this dataset is to measure the robustness of the model to different types of corruption it might encounter in the wild. The performance on a specific corruption is calculated as the mean over 5 different levels of perturbation strength. We list the corruption robustness for ResNet-18 in table 6. We notice that networks trained with TNA prove more robust to each of the corruptions, thus proving that the representations learned by TNA are stronger than those trained with regular training.

## 4.4 ABLATION STUDY

**The effect of $\alpha$.** By setting $\alpha = 0$, the logit matching loss term from $\mathcal{L}_{\text{TNA}}$ (eq. (1)) is removed, which in turn severs the dependency between the two networks during training and equates to training both independently. As such, varying the factor $\alpha$ results in more or less guidance via the twin network. To determine the optimal amount of guidance required, we perform an ablation test on the parameter $\alpha$, of which we list the results in table 7.

From these results it is clear that a higher $\alpha$, indicating more guidance of the twin network, results in better performance of the base network. However, at some point, a too large $\alpha$ can inhibit the training procedure, this is evident with MobileNetv2 + CIFAR-100, where if $\alpha = 0.1$ (last column of table 7) the model lags significantly behind the other settings in the first 50 epochs of training. An even larger $\alpha$ leads to non-convergence and networks that are no better than random chance.

**Differently sized twin networks.** As of now, in our experiments, we used twin networks of the same size as the base network. I.e., if the base network is a ResNet-18, then the twin network is a (differently initialized) ResNet-18. In this ablation, we remove that constraint, and study a combination of different ResNet models on the TinyImageNet dataset. We list the results in table 8. We notice that for ResNet-50 on this dataset, we require a lower value for $\alpha$ to avoid non-convergence. More specifically, for ResNet-50 we use $\alpha = 5E-4$ rather than $\alpha = 5E-3$ for the other settings.

Table 8: The impact of twin network size on base network accuracy for TinyImageNet.

| | Twin Network | | |
| --- | --- | --- | --- |
| Base Network | ResNet-18 | ResNet-34 | ResNet-50 |
| ResNet-18 | $64.75 \pm 0.47\%$ | $64.51 \pm 0.25\%$ | $62.17 \pm 0.53\%$ |
| ResNet-34 | $66.26 \pm 0.23\%$ | $65.97 \pm 0.41\%$ | $64.69 \pm 0.48\%$ |
| ResNet-50 | $65.08 \pm 0.38\%$ | $64.70 \pm 0.24\%$ | $64.94 \pm 0.77\%$ |

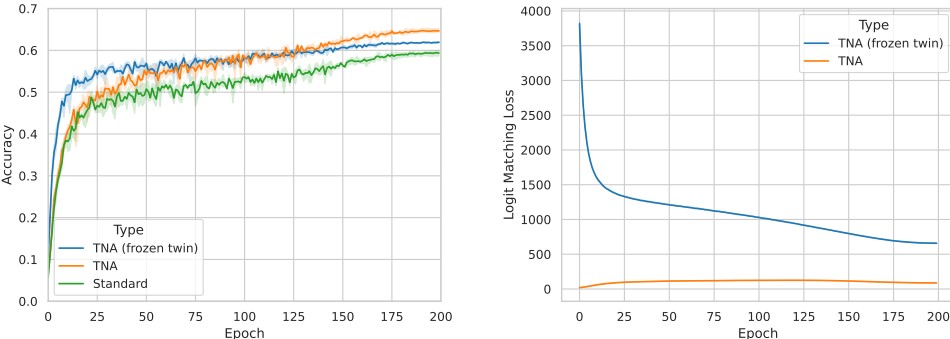

Figure 4: The impact of using a frozen fully-trained ResNet-18 model as twin on **(left)** TinyImageNet validation accuracy and **(right)** training loss.

We notice that applying TNA with a small and a large model produces a better performing large model, but a worse performing small model, than applying TNA with same size models. This shows that Logit Matching is seemingly harming the performance of the small model to benefit the larger model. We hypothesise that the cause of this lies in the different convergence behaviour of larger networks compared to smaller networks. To illustrate this, we have plotted the validation accuracy curves of standard training on TinyImageNet in fig. 3. We can see that during the initial epochs, the accuracy of the large ResNet-50 model is substantially lower, even though it converges to a higher final accuracy. When combining two networks with different convergence behaviours in TNA, this leads to an 'averaging' behaviour pulling down the smaller model. This phenomenon is not observed in traditional KD methods, in which the knowledge is extracted from a larger, fully trained model.

**Frozen, fully-trained twin networks.** One of the disadvantages of TNA is that training two networks in parallel is more computationally expensive than training a single network, as the computational graph is increased drastically. As such, a straightforward application would be to use a frozen fully-trained twin network as guidance for the base network. Not only would this vastly speed up the backward pass during training, the base network will already have access to high-quality logits at the start of training. This definition coincides with a Knowledge Distillation approach following Kim et al. (2021), in which the loss on the hard targets ($\mathcal{L}_{\text{CE}}$) is weighted significantly more than the loss on the soft targets ($\mathcal{L}_{\text{matching}}$).

In fig. 4 we use ResNet-18 + TinyImagenet to show the results. We notice that when using a frozen pretrained twin network, the validation accuracy of the base network is higher in the earlier epochs, but converges to a lower accuracy than the co-training of the twin network and the base network. We hypothesise that this is due to the concurrent mutual reinforcement of the twin and base network which provides better guidance within the later epochs than using a frozen network. This is evident in the evolution of the Logit Matching loss, where we notice that the logits of the base network and the frozen twin are substantially different.

**Late matching.** Another improvement, which has been proposed by Anil et al. (2018), is to apply the distillation loss in online KD in a later phase. While (Anil et al., 2018) only studied this with the KL Divergence, we also apply this on the L2 matching loss. The idea behind this approach is that two networks at initialization and during early training are too different, and as such matching

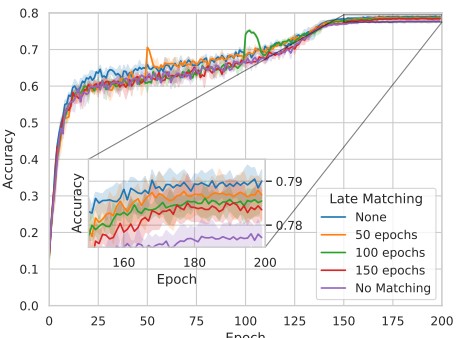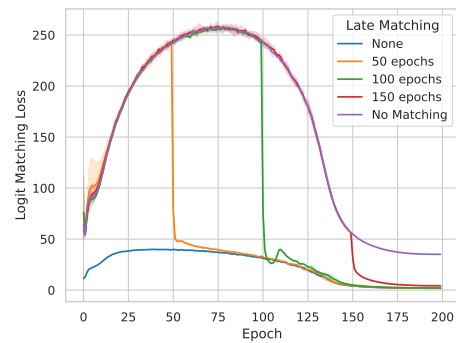

Figure 5: The accuracy (**left**) and logit matching loss (**right**) of ResNet-18 on CIFAR100 for different settings (at 0, 50, 100 and 150 epochs) of Late Matching.

the features would be counterproductive for their respective training. We conduct experiments for different starting points for the Logit Matching and show the results in fig. 5.

Analysing the results on ResNet-18 with CIFAR100, we notice that TNA without late matching outperforms the different late matching approaches, albeit with a small margin, and we link later matching with lower performance. Additionally, we notice that when late matching is activated, the validation accuracy jumps sharply in that epoch which coincides with a sharp drop in the logit matching loss, showing that the application of logit matching has an immediate and noticeable impact. Interestingly, this after a few epochs the validation accuracy drops again, albeit to a higher value. The cause of this is not further explored, but we hypothesise that there is likely a mismatch between the new position in the loss landscape and the current learning rate.

## 5   CONCLUSION

We have assessed the applicability of the Twin Network Augmentation methodology on a set of various CNN benchmarks, demonstrating that TNA consistently improves the baseline by significant margins. To determine the effect of the additional trainable parameters, we conducted an in depth study comparing TNA to similar approaches, and we show that for a given number of trainable parameters, TNA outperforms with an additional bonus of a lower inference cost. The performance gain can be boosted even further by ensembling the base and twin network at the cost of no longer having faster inference. Based on a set of carefully crafted ablations we discovered that the performance increase when training with TNA is linked to the matching of logits between two equally well-trained models, and that matching needs to occur throughout the training process for best results. Using a frozen model, or two models with too different sizes, results in less pronounced increases over the baseline, or even decreases.

Next, we conducted a thorough analysis of the representations learned through TNA to understand the underlying reasons for the superior performance of these networks compared to those trained via conventional methods. Our analysis reveals that although the low-level features are similar between standard trained networks and those learned via TNA, the main difference is made in the high-level features, which are more dissimilar to those of regularly trained networks. We link this difference to a higher utility in classification via linear probes.

**Next Steps.** While we show that TNA induces better high-level features within networks and as such, leads to better validation accuracy, we do not find a specific root cause for this phenomenon. As such, determining the exact cause for the observed better performance might allow training techniques with less overhead. Additionally, we can study different levels of supervision, such as matching intermediate features as opposed to matching only at the logit level.

ACKNOWLEDGEMENTS

This work is partially supported by the Dehousse Mandate id:54/RECBV007 from the Department of Computer Science of the University of Antwerp, the Flemish Government under the "Onderzoeksprogramma Artificiele Intelligentie (AI) Vlaanderen" programme and a SB grant (1S87022N) from the Research Foundation Flanders (FWO).

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

## A  TRAINING CONFIGURATIONS

### A.1  TNA AND STANDARD TRAINING

We list the training configurations used for each model in table 9. For each model, we used a SGD optimizer with the specified LR, except for the ViT for which we used ADAM.

Table 9: Training configurations used for different dataset network combinations.

| Network | Dataset | Batch Size | LR | Epochs |
|---|---|---|---|---|
| ConvNet | CIFAR-10 | 256 | 0.1 | 200 |
| ResNet-18 | CIFAR-10 | 256 | 0.1 | 200 |
| MobileNetv2 | CIFAR-10 | 256 | 0.1 | 200 |
| ConvNet | CIFAR-100 | 256 | 0.1 | 200 |
| ResNet-18 | CIFAR-100 | 256 | 0.1 | 200 |
| MobileNetv2 | CIFAR-100 | 256 | 0.05 | 200 |
| ConvNet | TinyImageNet | 256 | 0.1 | 200 |
| ResNet-18 | TinyImageNet | 256 | 0.1 | 200 |
| MobileNetv2 | TinyImageNet | 256 | 0.05 | 200 |
| ResNet-50 | ImageNet | 256 | 0.2 | 90 |
| ViT-S/16 | ImageNet | 1024 | 0.001 | 90 |

## A.2 NETWORK STRUCTURES

**ConvNet structure.** The ConvNet we use during our experiment with CIFAR-10, CIFAR-100 is listed in table 10. After each layer, we apply the ReLU activation function, and before the linear classification layer, we flatten the input. For TinyImageNet, we replace the last pooling layer with an adaptive pooling layer to ensure that the input to the linear layer remains the same shape.

Table 10: A simple ConvNet for use with CIFAR-10,

| Layer | Description |
|---|---|
| Conv1 | $3 \times 3$ Conv layer w/ 16 filters |
| Conv2 | $3 \times 3$ Conv layer w/ 32 filters |
| Pool | $2 \times 2$ Avg Pooling layer |
| Conv3 | $3 \times 3$ Conv layer w/ 64 filters |
| Conv4 | $3 \times 3$ Conv layer w/ 64 filters |
| Pool | $2 \times 2$ Avg Pooling layer |
| Fc | Linear layer w/ $1600 \times 10$ connections |

**ResNet-18 structure.** Following common practice, we modify the ResNet-18 structure for small-image datasets (CIFAR-10, CIFAR-100, TinyImageNet). We do this by replacing the first convolutional layer and omitting the first MaxPool layer. More specifically, rather than a convolutional layer with a $7 \times 7$ kernel, a padding of 3 and a stride of 2; instead we employ a convolutional layer with a $3 \times 3$ kernel, a padding of 1 and a stride of 1.

## A.3 LINEAR PROBING

To insert the linear probes, we first identified the residual blocks within a ResNet-18 model. Next, we freeze the weights up until that point and inserted a Downsampling layer via Average Pooling and a linear classification layer. By using this downsampling, we considered a single value per channel to avoid excessive computations.

Training the linear probes is done on the full training dataset with a reduced learning rate of 0.01 for 10 epochs. Afterwards the accuracy of the linear probe is calculated via the validation set, and this is considered as a proxy of the feature quality at that point in the network.

## B COMBINING TNA WITH DIFFERENT AUGMENTATION VIEWS

In the main paper, we have highlighted the similarities between TNA and Contrastive Learning. Both approaches attempt to align two views of the same datapoint. In the case of TNA, this is by

using feeding the datapoint through two distinct networks, while contrastive approaches instead feed two augmented views of the same datapoint through a single network.

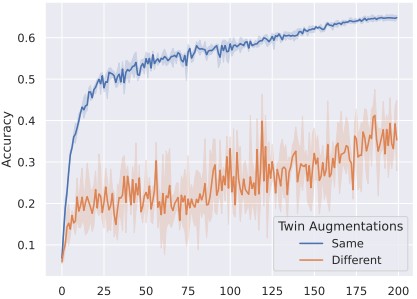 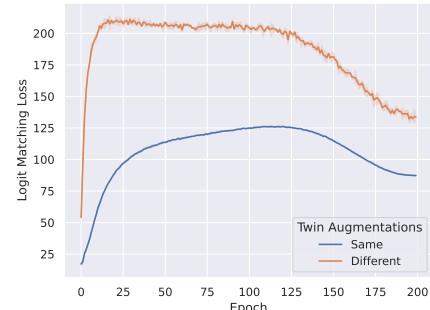

Figure 6: The convergence curves of the **(left)** validation accuracy and **(right)** Matching loss, for TNA on ResNet-18 + TinyImageNet with, and without different augment

While the effectiveness of both approaches has been shown, one could ask the question what would happen when combining both. This means generating two views of the same datapoint and then feeding each view through a distinct network, and matching the logits as such. In fig. 6, we show the convergence curves of TNA with and without differing augmentations.

We notice that the case where both networks receive different augmented views of the same sample, significantly impacts the convergence of the models. This is caused by a severely increased Logit Matching Loss, compared to the baseline TNA. Based on this empirical evidence, we hypothesize that the use of augmentations results in logits that are too different from each other, thus leading to an adverse effect on the training phase.

