# OpenReview forum: "Improving Neural Network Accuracy by Concurrently Training with a Twin Network"
_ICLR.cc/2025/Conference — ICLR 2025 Poster_

### Official Review · Reviewer_S9Mr · 2024-10-26

**Soundness:** 3
**Presentation:** 3
**Contribution:** 3
**Rating:** 8
**Confidence:** 3

**Summary:**

This paper investigates the potential use of Twin Network Augmentation (TNA), originally used to train spiking neural networks, in conventional CNNs trained for image classification in standard benchmarks such as CIFAR and ImageNet variants. The narrative then moves on to in-depth analyses of learning dynamics and ablation studies.

**Strengths:**

This paper is rigorous with a large number of convincing experiments and results. I appreciate the literature review of all similar methods and the summary provided in Figure 1 (even though it seems like they are a little too similar to have such drastically different names).

**Weaknesses:**

- Table 1 is quite chaotic. I can roughly understand why ResNet-18 is trained on TinyImageNet while ResNet-50 is trained on ImageNet and so on, but this is not rigorous or scientific, and can give the impression that the authors are simply cherry-picking results (I do not think this is the case). It would be better to keep one thing constant (network or dataset) and perform your experiments. (4 networks x 4 datasets is really only 16 cases, perhaps the authors can consider running them all).

- There seems to be a lot of attention placed on DML. Did the original TNA paper also compare DML and TNA for spiking networks? If so, any different observations found for spiking networks vs conventional CNNs? If not, why is so much analysis placed on DML? To be more specific, knowing that TNA is different from DML does not tell me anything more about TNA, which is the focus of this paper.

**Questions:**

- Did the original TNA paper talk about anything in Figure 1? Or is this an original attempt at unifying all the ideas?

- I might have missed this in the paper, but in the conclusion the authors claim that "the performance increase when training with TNA is linked to the matching of logits" but isn't matching the softmax pretty much just matching a non-linear function of the same quantity?

---

> ### Author Response · Authors · 2024-11-18
>
> Dear reviewer S9Mr,
>
> We thank you for your time spent reviewing our paper, here are our answers.
>
> **Chaotic Table 1.**
>
> Regarding table 1, our motivations are as follows. We have used the ConvNet as a proof-of-concept on the CIFAR-10 dataset, but have not applied this on other datasets, as this is quite a simple architecture, and not fit to achieve good accuracy on larger datasets. We do however agree that it is easier to have the same networks on different datasets, and that is why we have generated additional results for CIFAR100 and TinyImageNet. Unfortunately, in combination with other requests from reviewers, we do not have the computational resources to conduct additional experiments on ImageNet. Instead we have chosen for a recognisable baseline on ImageNet in the form of ResNet-50, as well as a Vision Transformer courtesy of reviewer 4gAE.
>
> **Comparison with DML.**
>
> The attention for DML is absent in the original TNA paper. Indeed, the reason for our attention on that comparison, is because this method is conceptually the closest to TNA, as it also involves training multiple models in parallel, but uses KL divergence to match probability distributions of the models, rather than L2 distance to match logit distributions. We believe that a thorough comparison with the closest method is desirable and something that was neglected in the original paper. By comparing directly, we show that L2 norm on the logits is a stronger criterion than KL divergence on the probabilities, similar as to [1] shows this for Knowledge Distillation.
>
> **Did the original TNA paper talk about anything in Figure 1? Or is this an original attempt at unifying all the ideas?**
>
> In the original paper, there is a figure that explains the functionality of TNA at training and inference, which is reasonably similar to Figure 1 (a), but they do not show visual diagrams of similar approaches. This figure serves as an easy visualization that positions TNA to other methods.
>
> **I  might have missed this in the paper, but in the conclusion the authors claim that "the performance increase when training with TNA is linked to the matching of logits" but isn't matching the softmax pretty much just matching a non-linear function of the same quantity?**
>
> Indeed, the probabilities are the result of a nonlinear function on the logits. However, we argue that a direct matching, without the nonlinearity results in a better performance, as we show from our results.
>
> We hope this answers some of your questions and considerations regarding the manuscript.
>
> Kind regards,
>
> The authors
>
> References:
> [1]: Taehyeon Kim, Jaehoon Oh, NakYil Kim, Sangwook Cho, and Se-Young Yun. “Comparing kullback-leibler divergence and mean squared error loss in knowledge distillation.”, arXiv:2105.08919

---

> > ### Comment · Reviewer_S9Mr · 2024-11-21
> > **Thank you**
> >
> > I thank the authors for the additional experiments in Table 1. Regarding the red values, perhaps the authors should not rush and take their time to give rigorous values (remember that rigor was the original intent of that request!) but I appreciate the effort and acknowledgement of my comment.
> >
> > The clarifications regarding the original TNA paper has been helpful.
> >
> > A brief glance at other reviews reveals that the additional training costs and novelty are common issues raised. I agree with the existence of both issues and I will it up to the rest of the reviewers to deem if they are addressed. Personally, the level of novelty has passed my own standards but it may not be the case for others.
> >
> > I will raise my score and wish the authors good luck.

---

> > > ### Author Response · Authors · 2024-11-22
> > >
> > > Dear Reviewer S9MR,
> > >
> > > We thank you for your appreciation of our work and our efforts.
> > >
> > > Kind regards,
> > > The authors

---

### Official Review · Reviewer_83Qm · 2024-10-29

**Soundness:** 3
**Presentation:** 4
**Contribution:** 2
**Rating:** 8
**Confidence:** 4

**Summary:**

The authors implement a new method called Twin Network Augmentation (TNA) for CNNs that improves the downstream performance by training two networks in parallel and using their logits in a combined loss function. The loss function consists of the cross entropy loss of each network's output with a given label and the mean squared error of the network's logits.
They compare their method with already established methods like Deep Mutual Training, as well as the standard machine learning approach with similar or larger networks (i.e. wide resnet compared to resnet with TNA).

**Strengths:**

The paper is written very well and easy to understand. The mathematical concepts are accessible and understandable. All the experiments align closely with the descriptions in the paper, effectively illustrating the theoretival insights. Notably, the paper includes a refreshing variety of experiments in the ablation studies, offering a comprehensive analysis of the TNA approach.

**Weaknesses:**

Major:
- Tables and figures could benefit from more detailed captions, specifying, for example, the network and dataset used. While all relevant information is present in the text, including it in the captions would enhance clarity and accessibility.
- The rationale behind the choice of data sets for specific instances is not always clear.

Minor:
- Tables 3 and 4 appear to be very small.
- For Figure 2, a bar plot may be more suitable.
- Table 7 may be improved by introducing a column with an alpha value of 0.

**Questions:**

1. In 3.3, describing the network architectures (i.e. ResNet-18, ResNet-50,...) and corresponding datasets: Usually, for small images (e.g. 64x64 in Tiny ImageNet) the first 7x7-conv-layer of a ResNet is swapped for a smaller 3x3-conv-layer without stride. Did you follow this protocol?
2. Tables 1 and 2 use the terms 'Default Train' and 'Single Train,' respectively. Is there a distinction between the two?
3. Both the base and twin networks currently receive the same image. Have you considered applying slightly different augmentations to each network, as is commonly done in contrastive learning approaches like SimCLR (https://arxiv.org/abs/2002.05709)?
4. In 4.4.1, why does a larger alpha result in networks that perform no better than random chance? High values of alpha (much greater than 1) inherently produce larger gradients, necessitating an adjustment in the learning rate. Could this be a contributing factor to poor performance?
5. In section 4.4.4, you investigated late matching. Have you thought about 'warming up' the alpha (e.g. exponentially)?
6. In the right-side plot of Figure 5, the curve for late matching at epoch 150 appears to decline independently, even before matching. What does the logit matching loss look like in the absence of any matching?

---

> ### Author Response · Authors · 2024-11-18
>
> Dear reviewer 83Qm,
>
> We thank you for your time spent reviewing our paper. To improve clarity, we have rewritten parts of the paper, including the captions, where possible we have incorporated your suggestions regarding tables and figures, and we thank you for those suggestions.
>
> **Rationale behind dataset choices.**
>
> Regarding the choice of datasets, we agree that this was sometimes murky. To accommodate for this issue, we have added experiments with ConvNet on CIFAR-100, TinyImageNet, and MobileNet-v2 on TinyImageNet. This means that for CIFAR-10, CIFAR-100, TinyImageNet, all three networks are evaluated. Additionally, we have added a Vision Transformer trained on ImageNet to highlight the effectiveness of TNA on more recent approaches. For the in-depth experiments, we have opted to list results for ResNet-18 and TinyImageNet. The reason is that this is the most complex baseline we employed, for which we had the computational resources to do an in-depth study. Additionally, the ResNet-18 model is well-suited for comparisons with WideResNet, which is not the case for ConvNet or MobileNet-v2.
>
> **1. In 3.3, describing the network architectures (i.e. ResNet-18, ResNet-50,...) and corresponding datasets: Usually, for small images (e.g. 64x64 in Tiny ImageNet) the first 7x7-conv-layer of a ResNet is swapped for a smaller 3x3-conv-layer without stride. Did you follow this protocol?**
>
> The ResNet-18 we employ for the smaller resolution datasets is indeed modified as described, we failed to mention this in the paper, and have added this to the appendix.
>
> **2. Tables 1 and 2 use the terms 'Default Train' and 'Single Train,' respectively. Is there a distinction between the two?**
>
> ‘Default Train’ and ‘Single Train’ were used interchangeably in the paper, but we do concur that a single term gives more clarity. To this end, we have chosen the term ‘single train’ to employ in the paper, as this also succinctly describes the training process, and have revised the paper accordingly.
>
> **3. Both the base and twin networks currently receive the same image. Have you considered applying slightly different augmentations to each network, as is commonly done in contrastive learning approaches like SimCLR?**
>
> We have considered increasing representational variety with transformations such as SimCLR employs, but have decided to not implement them in the main paper, as we wanted to decouple the effect of different representations via different networks from that of different representations via different augmentations. We hypothesize that this might hinder the convergence process, as different augmented views with different networks might elicit wildly varying logits. That being said, we feel that it is a valuable addition to the paper, and we will describe the results of that experiment in the appendix.
>
> **4. In 4.4.1, why does a larger alpha result in networks that perform no better than random chance? High values of alpha (much greater than 1) inherently produce larger gradients, necessitating an adjustment in the learning rate. Could this be a contributing factor to poor performance?**
>
> A higher value of alpha will shift the trade-off between optimizing for classification accuracy and representation similarity too much. We have observed that in that case the optimization process will shift to optimizing the feature similarity earlier, and foregoes improvements in the classification loss until later. The convergence to random chance can be influenced by a high learning rate, and as such can be ameliorated by reducing the learning rate.
>
> **5. ... Have you thought about 'warming up' the alpha (e.g. exponentially)?**
>
> A warmup schedule for alpha with late rewinding has not been considered. We felt that additional complexity we would introduce with the combination of the type of warmup schedule (linear, exponential, sigmoid, …) and the starting, and ending points of the schedule, would be too computationally expensive for an in depth analysis. We will leave this for further analysis, as this could yield additional benefits.
>
> **6. In the right-side plot of Figure 5, the curve for late matching at epoch 150 appears to decline independently, even before matching. What does the logit matching loss look like in the absence of any matching?**
>
> Regarding figure 5, we have not directly logged the Logit Matching Loss when not considering TNA. However, for the purpose of comparison, we have generated these results and noticed that this curve indeed follows a parabolic path. However, without considering the loss term, the L2 distance between logits of different networks  converges to a value far higher than with the loss term. We have added the figure in the updated manuscript. We should note that these results have not been observed for ResNet-18 on TinyImageNet, and as such are likely an artifact of the CIFAR100 dataset.
>
> We hope this resolves some of your questions and considerations.
>
> Kind regards,
>
> The authors

---

> > ### Comment · Reviewer_83Qm · 2024-11-25
> > **Thank you**
> >
> > Thank you for addressing my comments in the revised version of your manuscript! I highly appreciate the effort you put into improving the clarity and quality of the work. I have adjusted my evaluation accordingly and wish you all the best in the subsequent stages of the review process.

---

> ### Author Response · Authors · 2024-11-25
>
> Dear reviewer 83Qm,
>
> Thank you very much for the appreciation of our work and our efforts. Without a doubt, your feedback also helped us improving our manuscript that much more.
>
> Kind regards,
>
> The authors.

---

### Official Review · Reviewer_BRu9 · 2024-10-31

**Soundness:** 2
**Presentation:** 3
**Contribution:** 2
**Rating:** 5
**Confidence:** 3

**Summary:**

This paper first introduce Twin Network Augmentation (TNA) from SNN, then validate the viability of this method on a wide range of popular CNN benchmarks and compare this approach to existing Knowledge Distillation schemes.

**Strengths:**

This paper introduces the TNA method from the SNN field into CNNs for the first time, demonstrating its applicability across a variety of CNN models. The TNA method improves overall performance by simultaneously optimizing two networks and using MSE loss to adjust their logits.

**Weaknesses:**

1.This paper validates the feasibility of the TNA method in the CNN field. Although the experiments in this paper are comprehensive, they do not sufficiently analyze the reasons why the TNA method is feasible. Considering that this paper does not propose a new method, I doubt whether the work meets the standards of the ICLR conference.

2.Could the author explain why TNA is superior to DML? According to Tables 2 and 7, TNA only surpasses DML under special alpha values.

3.TNA requires training two networks simultaneously, resulting in doubled training costs. How to balance these costs against performance gains?

4.Section 4.3 and Table 5 are confusing. Please clarify what each column in Table 5 represents.

**Questions:**

Why does Eqn. 2 not align with the DML loss described in Figure 1 (b)?

---

> ### Author Response · Authors · 2024-11-18
>
> Dear Reviewer BRu9,
>
> We thank you for the time spent reviewing our paper. Below, we will address the weaknesses and questions raised by your review.
>
> **1.** We argue that the introduction of a totally new method is not the only criterion for novelty.  While [1] introduced the TNA method, it was validated on a single small model, trained on relatively easy vision datasets. Additionally, comparative study with different approaches was limited to a single experiment with knowledge distillation. In contrast, we show that this method is also applicable on the more commonly used paradigm of CNNs. This statement is well supported with experiments on numerous benchmarks, including the commonly–used ImageNet dataset (See Table 1). To further complement these results, we compare against methodologies that employ a similar number of trainable parameters (Table 2, 3, 4), and demonstrate that TNA is more effective, while requiring less computational overhead in the inference phase. Finally, we demonstrate how TNA influences the features that are learned in the networks.  These are all aspects that are not touched upon in [1] and are valuable for understanding the applicability of TNA.
>
>
> **2.** The alpha is a trade-off parameter for the Logit Matching Loss. DML introduced no such parameter. Indeed, we do not claim that for every possible value of alpha TNA outperforms DML. Additionally, by using this alpha, we can negate effects of the additional loss term, and avoid modifying the learning rates negatively. An example can be found in Table 2, where DML results in exploding gradients and non-convergence at lr=0.1, which is the more optimal choice for training a single ResNet-18 model on TinyImageNet. TNA has no such issue, as we can control the influence of the additional loss term.
>
>
> **3.** We show in Table 4 that the accuracy gain of training two models at the same time is higher than training a single model with double the parameters. This means that it is more efficient to concurrently train, rather than adding additional parameters. Additionally, this cost is only in the training phase, while in the inference phase, we require less computation than the double parameter models. To further increase the accuracy gain of training two models, we can additionally ensemble both models, and we show in Table 3, that this results in additional performance gains, at the cost of the reduced inference time.
>
> **4.** This has been resolved in the new version.
>
>
> **Why does Eqn. 2 not align with the DML loss described in Figure 1 (b)?**
>
> The DML loss in Eqn. 2 is related to training model A, in Figure 1 (b), we show the full loss for training both model A & B with DML.
>
> We hope this answers your questions and makes you reconsider some of the weaknesses.
>
> Kind regards,
>
> The authors

---

> ### Comment · Reviewer_BRu9 · 2024-11-23
>
> I appreciate the authors' response, as it has addressed some of the concerns I raised.
> Based on their efforts, I have raised my score to 5.
> But I believe the experiments and analysis of this paper are still insufficient.
> This work should be categorized within the field of self-distillation, and the benchmark used for comparison, DML, is a paper from six years ago.
> I suggest that the authors compare with more recent self-distillation papers such as [1][2][3][4], etc., and involve more types of task in experiments.
>
> [1] Zhang L, Song J, Gao A, et al. Be your own teacher: Improve the performance of convolutional neural networks via self distillation[C]//Proceedings of the IEEE/CVF international conference on computer vision. 2019: 3713-3722.
>
> [2] Wu L, Li J, Wang Y, et al. R-drop: Regularized dropout for neural networks[J]. Advances in Neural Information Processing Systems, 2021, 34: 10890-10905.
>
> [3] Zhang C B, Jiang P T, Hou Q, et al. Delving deep into label smoothing[J]. IEEE Transactions on Image Processing, 2021, 30: 5984-5996.
>
> [4] Yang Z, Zeng A, Li Z, et al. From knowledge distillation to self-knowledge distillation: A unified approach with normalized loss and customized soft labels[C]//Proceedings of the IEEE/CVF International Conference on Computer Vision. 2023: 17185-17194.

---

> > ### Author Response · Authors · 2024-11-25
> >
> > Dear reviewer BRu9,
> >
> > We thank you for appreciation. We have thoroughly read the included references, and while we notice some similarities with our proposed methodology -- such as the regularized dropout -- we feel that there is some distinct difference as self-distillation uses the information at different levels from within the same model, while we use the information from another model. That being said, in future research we will most definitely consider these methodologies for comparison. However, in the light of the coming deadline and the lack of computational resources, we unfortunately cannot provide these results for this rebuttal phase.
> >
> > Kind regards,
> >
> > The authors

---

### Official Review · Reviewer_4gAE · 2024-11-01

**Soundness:** 2
**Presentation:** 2
**Contribution:** 2
**Rating:** 5
**Confidence:** 4

**Summary:**

This paper investigates the Twin Network Augmentation (TNA) method, originally proposed for Spiking Neural Networks, and evaluates its effectiveness on Convolutional Neural Networks (CNNs). The authors show that concurrently training two networks and matching their logits using Mean Squared Error (MSE) loss improves validation accuracy across various CNN benchmarks. They compare TNA to traditional Knowledge Distillation methods, highlighting its superior performance and robustness.

**Strengths:**

This paper conducts extensive experiments to demonstrate that training two twin networks with the same objective, while simply minimizing the mean squared error of their output logits, can significantly enhance the performance of a single network. The primary contribution of this paper lies in validating the effectiveness of this approach not only for SNNs but also for CNNs.

**Weaknesses:**

1. This paper simply transfers the method from reference [1] to traditional CNNs, showing some performance improvement based on the experimental results. Although the authors conducted a wide range of experiments from different angles to validate its effectiveness, the overall contribution lacks novelty. In other words, the only modification was replacing SNNs with CNNs, which is trivial.

2. This method is only validated on the classification task and is only compared with the outdated backbones.

*Reference*

[1] Deckers, L. et al. Twin Network Augmentation: A Novel Training Strategy for Improved Spiking Neural Networks and Efficient Weight Quantization.

**Questions:**

1. The application of mean squared error (MSE) loss on logits enhances performance but doubles the training parameters when using twin networks like ResNet-18. It is essential to assess whether ResNet-18 serves as a reasonable baseline, especially in comparison with models that have similar parameter counts. The objective should be to establish a computationally feasible baseline.

2. Although Section 4.2 offers some intuition for the performance improvements, a more rigorous theoretical analysis is required to enhance understanding of the underlying mechanisms.

3. The writing of this paper should be largely improved. For example, lines 58 and 187 must utilize `\citep` for proper citation formatting. Additionally, both Figure 5 and Table 10 are missing concluding periods, and line 630 must also end with a period for consistency.

---

> ### Author Response · Authors · 2024-11-18
>
> Dear reviewer 4gAE,
>
> We thank you for your time spent reviewing our manuscript. Below, we will address your weaknesses and answer your questions.
>
> **Lack of Novelty.**
> We respectfully disagree with your assessment that replacing SNNs with CNNs is a trivial modification. Indeed, purely from a plug-and-play view, this is accurate, but this disregards many factors. SNNs and CNNs are fundamentally different architectures. While a classical CNN, or indeed any of the ANN models (also including Recurrent networks, Transformers, Fully connected networks, …) function with stateless neurons on real-valued inputs and outputs, the SNN instead employs stateful neurons and uses binary spike trains as input and output. This has the implication that SNNs have different training dynamics, which are incurred by the encoding of spike trains, and the additional temporal dynamics, which can cause the SNN to learn features in a different manner than ANNs. Furthermore, due to the limitations of SNNs, typical deep networks are difficult to train, such as larger ResNets, which is exemplified in [1], where the authors employ a relatively shallow network with depth 8.
>
> This in combination with the limited experiments in [1], focusing only on simple datasets, and generally not conducting exhaustive comparisons on the impact of parameter count, together with an in-depth analysis of the features, leads us to believe that our contribution does have a merit. While we are not introducing a new method, instead we show its feasibility on a different neuronal paradigm and support this with rigorous testing on several benchmarks, thus showing its effectiveness in non-trivial settings.
>
> **TNA is only validated on classification methods.**
> In theory TNA could also be applied to other domains such as object detection, as the additional loss term only requires logits to match between different models. However, we chose to limit our investigation to the classification problem, as this is the domain considered in the introducing paper. That being said, if time and computational resources allow us, we will conduct a benchmark comparison on the object detection domain during the rebuttal period, and describe it in the appendix.
>
> **TNA is only validated on outdated models.**
> The critique of only outdated models is in our opinion reasonable. Due to computational restrictions, we chose commonly used benchmarks in the form of residual networks. However, to accommodate this critique, we have conducted an additional experiment with Vision Transformers on ImageNet, where we largely follow the training plan of [2] (excluding advanced augmentations). This shows the applicability of TNA on more recent approaches.
>
> **1. The application of mean squared error (MSE) loss on logits enhances performance but doubles the training parameters when using twin networks like ResNet-18. It is essential to assess whether ResNet-18 serves as a reasonable baseline, especially in comparison with models that have similar parameter counts. The objective should be to establish a computationally feasible baseline.**
>
> We have in fact conducted experiments related to parameter count. This involved comparing a TNA with two ResNet-18 models to a single WideResNet-18-2 in Table 4. This allowed us to demonstrate that the accuracy gain of TNA is higher than that of training a model with double the parameters. Additionally, we have shown that we can also outperform model ensembling in Table 3, which is another technique to increase the effective capacity of a neural network. We have validated TNA on a number of different models ranging from small (ConvNet) to large (ResNet-50). In addition to those models, we have also added a Vision Transformer in the rebuttal, and demonstrate the applicability on that as well.
>
> **2. Although Section 4.2 offers some intuition for the performance improvements, a more rigorous theoretical analysis is required to enhance understanding of the underlying mechanisms.**
>
> Unfortunately, we feel that we cannot adequately accommodate this request. We will notice that the similar method of DML [3] also offers no theoretical motivations.
>
> **Quality of writing.**
>
> Regarding the quality of writing of the paper, we have gone through the paper and carefully revised the text.
>
>
>
> We hope these answers satisfy your questions and help rethink some of the weaknesses of the paper.
>
> Kind regards,
>
> the authors
>
> References:
>
> [1] Deckers, L. et al. Twin Network Augmentation: A Novel Training Strategy for Improved Spiking Neural Networks and Efficient Weight Quantization.
>
> [2] Beyer, L. et al. Better plain vit baselines for imagenet-1k
>
> [3] Zhang, Y. et al. Deep Mutual Learning

---

> > ### Author Response · Authors · 2024-11-25
> >
> > Dear reviewer 4gAE,
> >
> > We would appreciate it if a response could be provided to our rebuttal. This will allow us to foster a nice discussion and can provide us additional feedback for our research. This in turn could be translated, if time allows, in added improvements to the manuscript before the rebuttal deadline.
> >
> > If the reviewer feels we have not sufficiently addressed their concerns, we would kindly appreciate some targeted feedback. Otherwise, we would be grateful if the reviewer considered increasing their score.
> > Kind regards,
> >
> > the authors

---

> > > ### Comment · Reviewer_4gAE · 2024-11-27
> > > **Official Comment by Reviewer 4gAE**
> > >
> > > Dear authors,
> > >
> > > Thank you for your response.
> > >
> > > I believe the authors' rebuttal addressed some of my concerns, but not all of them.
> > >
> > > *The unresolved issues:*
> > >
> > > Firstly, I find the explanation regarding novelty insufficient. I am familiar with many aspects of SNNs, including their training and principles. The authors spent considerable effort discussing the differences between SNNs and CNNs and mentioned that training SNNs is more difficult than training CNNs. However, this only emphasizes that applying SNN-based methods to CNNs represents a simplification of the task, not an innovation.
> > >
> > > Secondly, as the authors mention, TNA can be applied to other complex downstream tasks. However, testing it only on classification tasks does not provide a strong foundation for its generalizability or robustness.
> > >
> > > Finally, regarding the theoretical aspects, I understand that the authors wish to introduce a new model from an experimental perspective. However, as this is an ICLR submission, I believe offering theoretical insights or suggestions would be highly constructive and could significantly enhance the robustness of the work.
> > >
> > > *Improvements made:*
> > >
> > > The authors added experiments with Vision Transformer. Although this model is not new, it serves as a useful baseline. Additionally, the experiments with WideResNet-18 demonstrate that TNA is more effective than introducing a double parameter count. I believe this experiment can be considered one of the key demonstrations of TNA’s effectiveness.
> > >
> > > In summary, I appreciate the authors' rebuttal and respect their efforts in providing additional experiments. Adding extra experiments during the rebuttal phase is a difficult and challenging task. As such, I have made an effort to increase my score. However, as mentioned earlier, I still believe there are unresolved issues that cannot be addressed in the current version of the paper. I hope the authors understand my evaluation and I sincerely look forward to the development of an improved TNA model in future versions.
> > >
> > > Best,
> > >
> > > Reviewer 4gAE

---

> > > > ### Author Response · Authors · 2024-11-29
> > > >
> > > > Dear reviewer 4gAE,
> > > >
> > > > We thank you for your thorough response and your appreciation of our efforts. We will certainly incorporate your suggestions in the next phases of this research.
> > > >
> > > > Kind regards,
> > > > the authors

---

### Author Response · Authors · 2024-11-22

Dear reviewers,

We have uploaded a modified version of our manuscript in which we have addressed common weaknesses in the paper. Additionally, we have added a Vision Transformer baseline, to serve as an application on a modern architecture, and added experiments with different versions of data augmentation in the appendix. If any concerns remain after the rebuttals and the modified versions, please let us know.
If your concerns have been addressed, we would appreciate it if you could increase your initial scores to reflect upon this.

Kind regards,

the authors

---

### Meta-Review · Area_Chair_AmKm · 2024-12-26

**Metareview:**

the paper investigates the application of Twin Network Augmentation (TNA), originally proposed for spiking networks, to CNNs. The key claim is that training two networks concurrently while matching their logits via mse, improves validation accuracy across various CNN architectures and benchmarks

positive:
- good empirical validation across multiple architectures (ConvNet, ResNet-18/50, Vision Transformer) and datasets (CIFAR, TinyImageNet, ImageNet)
- good ablation studies examining various components of TNA
- demonstration that TNA's benefits extend beyond just increased parameter count

weaknesses:

- limited theoretical analysis explaining why TNA works better than alternatives
- while extensive, experiments mostly use traditional architectures rather than cutting-edge models
- no comparison with recent self-distillation methods
- the computational cost implications of using two networks are not fully discussed.

The authors managed to satisfy reviewers' concerns, thought the paper is borderline. I vote for accepting the paper.

**Additional Comments On Reviewer Discussion:**

The discussions were around the following point :

- novelty concerns: Authors argued that applying TNA to CNNs is non-trivial due to fundamental differences between SNNs and CNNs, though some reviewers remained unconvinced.
- multiple reviewers requested stronger theoretical analysis, which remains an open point. (though all reviewers increased their score)
- authors added Vision Transformer experiments to address concerns about dated architectures.
- authors demonstrated TNA's superiority over simply using larger models through WideResNet comparisons.

Most concerns were addressed through additional experiments or clarifications, though some theoretical aspects remain unexplored.

---

### Decision · Program_Chairs · 2025-01-22

Accept (Poster)